# Disruptive Innovation in Traditional Clusters: The Case of the Kerajet Ceramic Tile Cluster in Spain

**Jose Albors-Garrigos [1,*]** and **Jose Luis Hervas-Oliver [1,2]**

1  Departamento de Organización de Empresas, Universitat Politècnica de València, 46022 Valencia, Spain; jose.hervas@omp.upv.es
2  Departamento de Ingeniería, Universidad de la Costa, Cl. 58 ##55 - 66, Barranquilla 080020, Colombia
*  Correspondence: jalbors@doe.upv.es; Tel.: +34-646748455

**Featured Application: Public innovation policy makers, clusters agents.**

**Abstract:** Academic literature has often emphasized how firms in regional clusters exploit both place-specific local resources and external, world-class knowledge to strengthen their competitiveness by expanding the influence of regional systems of innovation. Innovation based on more complex and disruptive technologies tends to also be based on more open systems that utilize the clusters' external networks. However, most of the literature has associated clusters with incremental innovation. This paper will analyze the determinants of disruptive innovation development in traditional (low and medium tech) clusters caused by high-tech entrepreneurs. It will analyze the case of the development of breakthrough innovation, its diffusion in the Spanish ceramic tile cluster, and its consequent diffusion in the industry worldwide. It will examine how market demands, customer orientation, technology diffusion from other industries, industry competitiveness, as well as internal and external networking of clusters can facilitate the development of complex technology within a common set of social capital goals, cognitive schemes, and knowledge. The paper is based on a case study and field work carried out over10 years in the field in the Italian and Spanish tile ceramic clusters. The main contribution of this paper to technology strategy theory will be thorough the utilization of the disruptive technology paradigm in explaining industry changes and sustainability.

**Keywords:** tile ceramic clusters; low and medium tech clusters; radical innovation; disruptive technology

## 1. Introduction

Some of the works in the academic literature on this topic have outlined the industrial specialized approach of clusters versus the wider approach of regional innovation systems [1,2]. Firms in regional clusters exploit both place-specific local resources as well as external, world-class knowledge to strengthen their competitiveness by expanding the influence of regional systems of innovation [3]. The local "buzz" that clusters can create is also analyzed in order to understand cluster competitiveness and sustainability [4]. Clusters are also genuine "open innovation" spaces where knowledge creation and diffusion becomes stronger. Nevertheless, some of the literature has also pointed out how specialized knowledge circulation seems to be limited to certain communities within clusters as a result of the specialization of suppliers and actors [5]. Belussi and Sedita [6] have posited how clusters can develop learning dynamics though two alternative mechanisms: by exploiting local cluster knowledge or exploring and acquiring knowledge from external and distant agents.

In general, academic literature has associated clusters with incremental innovation [1,5,7,8], especially innovation related to traditional industries [9,10]. However, recent research concluded that clusters can provide a suitable environment for radical innovations [11]. Thus, there is a gap in relation

to the outcome of breakthrough technology and radical innovation in clusters. We need to ask if this is really possible, or whether it is, in fact, due to the effects of serendipity.

The object of this paper is to analyze the determinants of disruptive technology (or radical innovation) development in low- and medium-tec or mature industries within a cluster context. We leave the study of clusters' features to others and focus on a fertile cross-field study of clusters as frameworks that support and foster rapid breakthroughs, creation, and diffusion.

There are several research questions that our research will try to answer: Is it possible for a traditional industry based on a cluster to evolve radically? How is change induced—internally or externally? Which type of knowledge between tacit and explicit is predominant? How can disruptive innovation develop within a cluster? Who are the change promoting agents? Which barriers must be overcome and are there facilitating factors within the industry context that play an essential role?

The case of the development of digital printing technology has been analyzed, a global breakthrough event in the decoration phases of the ceramic tile process that has taken place in the Spanish ceramic tile cluster.

The paper will consist of an analysis of the development of digital printing and decorating technology, a global breakthrough in the decoration phase of the ceramic tile process that took place in the Spanish ceramic tile cluster between 2000 and 2010. Key factors will be examined, such as the evolution of the technology, the demands of the market, customer orientation, globalization and technology diffusion, and competitiveness from other industries. There will also be an analysis of how internal clusters and external networking facilitated the development of sophisticated digital technology, despite the existence of a predominant mechanical culture, a low level of training in information technology, and the maturity of the industry. There is also an investigation of the effects of the disruptive technology paradigm. This approach has been advantageous in explaining the evolution of this technology diffusion, which has overcome the inhibiting cluster barriers affecting the ceramic tile industry worldwide.

The paper begins with a description of the research methodology, and then introduces the reader to the context of the European ceramic tile industry. It follows with an analysis of the theory related to clusters and disruptive innovation. A case study is then analyzed. The paper concludes by discussing the findings and their implications.

## 2. Research Methodology

The paper is based on a case study of a small company, Kerajet, which is based in Castellón in the heart of the ceramic tile cluster. This company was founded by two electronic engineers and a chemist with a background and ample experience in mechanical engineering, the ceramic tile process, and information technology. These entrepreneurs had a clear vision of the digital future of ceramic tile decoration that could facilitate both the introduction of just-in-time production in the ceramic tile processes and the development of *avant garde* decoration designs. The introduction of ink jet tile decoration involves a process of radical innovation, which is changing the industry and its value chain. We conducted more than twenty formal and informal interviews with the various firms involved in order to fully understand the new technology and its impact on the cluster. Additionally, we analyzed the critical process of diffusion and adoption by different groups of users in the cluster. Additionally, the paper is based on more than ten years field study of the Spanish and Italian ceramic tile clusters, as well as participation in a European Commission project between 2002 and 2005.

This case will shed some light on the development of radical innovation in terms of its causes, barriers, and facilitating factors. According to Yin [12], a single case analysis can be useful as a critical test for existing theory. It can provide a holistic approach by analyzing larger units. Furthermore, the case study allows for the identification of key research questions and a clear rationale for discussion [13]. This technique is justified by the large size of the cluster, its market dimension, and the singularity of the ethnographic strategy followed [14].

The data used was based on the information collected through numerous interviews held with managers and engineers from suppliers, producers, distributors, and other actors in the Spanish and Italian ceramic tile clusters. The information was collected during visits to the ceramic tile exhibitions at CERSAI (International Exhibition of Ceramic Tile and Bathroom Furnishings) in Bologna and at CEVISAMA (International Fair of Ceramic Tiles and Bathroom Furnishing) in Valencia during 2000–2015. Also, we exploited the secondary information from industry reports and other available material.

## 3. The Ceramic Tile Production in Europe

### 3.1. An Overview

The world ceramic tile industry is clustered and connected through a global value chain. Within this, knowledge is created and disseminated mainly by the European clusters and their respective knowledge-based suppliers, namely glazing (chemicals) suppliers and machinery suppliers in a supplier-driven innovation pattern [15]. The clusters of Italy and Spain accounted for 15.4% of the world's export share in 2018 [16]. However, this proportion is being eroded continuously due to the spectacular growth of the new players, namely Brazil, Mexico, and China.

More than 300 firms compose the Castellón cluster in Spain from various related industries located nearby (ceramic producers, glazing suppliers, clay providers, ceramic machinery suppliers, transport agencies, distributors, among others). It produces roughly half of the ceramic tiles in Europe [17,18].

Along with Spain, Italy has one of the largest ceramic industries in the world. Around 83% of Italy's ceramic tile production (6.152 billion square meters) is concentrated in the Emilia-Romagna region in the Sassuolo area, where some 19,692 direct ceramic tile production jobs are located [19]. In addition, the Sassuolo area hosts a further 6000 jobs in the supplier industries, while Castellón has a production output of 531 million square meters and 15,400 direct jobs.

The Italian district includes a series of public organizations, such as the Centre of Ceramics of Bologna (CCB), which is analogous to the Technological Ceramic Institute in Castellón (ITC), and private bodies such as the industry trade association, the Assopiastrelle (whose Spanish equivalent is ASCER (Asociación Española de Fabricantes de Azulejos y Pavimentos Cerámicos). These bodies connect all the regional ceramic tile industries. Each of the two countries accounts for 9.3% of worldwide ceramic tile production, while Asia accounts for 70% [15,18].

### 3.2. The Role of Clusters

The world ceramic tile industry follows a pattern of major equipment innovations coming from the Italian ceramic tile equipment industry, which has expanded with direct investments being made in Brazil, Spain, and China. Similarly, worldwide innovations in glazing and pigments (in the chemical industry) have come from the Spanish ceramic tile glazing industry, which has also expanded worldwide. Therefore, the Italian and Spanish ceramic local value chains have each led a specific part of the global value chain (GVC) and are interdependent and related [20]. Auxiliary industries in this GVC also have different compositions in Spain and Italy, where there are different technological regimes [20,21] in each cluster. A technological regime can be broadly defined as "the particular combination of four fundamental factors: technological opportunities, appropriability of innovation, cumulativeness of technological advances, properties of the knowledge base" [22]. Thus, the strength of the Italian district is based on a strong base of machinery manufacturing specific to the ceramic tile industry, while the core asset of the Castellón cluster is a world-class glazing industry, confirming the cluster labor division posed by some authors [23]. These firms in the auxiliary industries lead the world and offer their innovations to Italian and Spanish ceramic tile manufacturers well before other clusters can acquire and assimilate the new technologies. Besides, these two interconnected cross-cluster industries are responsible for the majority of breakthrough innovations in the GVC. The process [18] shows the reciprocal links between ceramic tile clusters in Castellón (Spain) and Emilia-Romagna (Italy).

The tacit knowledge gained from operations and activities in the Castellón cluster is generated through interaction with local industries and various local organizations, and this knowledge is then partially transferred to the Italian cluster. The tacit knowledge learned in Castellón is, therefore, disseminated. At the same time, it is also complemented with the knowledge acquired from those other clusters in which the glazing subsidiaries operate, since the internal MNE (Multi National Enterprises) network knowledge is not sufficient to cope with conditions in different clusters, such as the Italian cluster. Similarly, the Italian equipment firms also create tacit knowledge in the Emilian cluster through interfirm interactions, and transfer part of this knowledge to the Castellón cluster. However, in dealing with local realities and the unique circumstances in Castellón, the necessary knowledge is obtained through interaction with local glazing firms and ceramic tile producers, combining local knowledge with partially complementary explicit knowledge from the Emilian cluster. In this way, new tacit knowledge is created. MNE internal networks cannot supply the local knowledge necessary for Italian equipment manufacturers to operate in Castellón using the red-limestone production process. Simultaneously, part of the knowledge learnt in Castellón concerning the red-limestone process is also disseminated to the Emilian cluster and other worldwide clusters via MNE subsidiary internal networks. Nevertheless, this study did not analyze the rest of the global value chain, as the dynamics described form part of a set of incentives that encourage innovation in both sectors, provoking a change in the structure of the ceramic global value chain [17].

Both Italian and Spanish clusters are well prepared organizationally, having sufficient public and private mechanisms to provide proper support to the value chain [17,24,25]. However, the process of interaction with organizations such as the ITC and their collaboration with the Jaume I Universitat seem to work better in Castellón [24]. Nevertheless, the real strength of the Castellón cluster lies in its systemic behavior, a mechanism of innovation diffusion that operates in a manner that is very difficult to replicate in a different context, as confirmed in interviews carried out while preparing this paper. Ceramic tile company technicians are in continuous contact with technicians from glazing companies. At the same time, ceramic tile companies hire chemical engineers specializing in ceramic tiles who were trained at the ITC and the Jaume I Universitat. Accordingly, there is a dynamic information and knowledge flow within the cluster network system. This flow is why the glazing industry is the primary signatory of contracts with the ITC and is the cluster sector with the most developed R&D. Knowledge is transferred through its interrelations and links with tile companies. At the same time, these links are strengthened by the ITC's support of the tile companies and by the hiring of technicians experienced in the various industries. These links create a fluid circulation of tacit and explicit knowledge. This process is aided by the use of a common language, culture, understanding, and by the personal relationships of the local workers who are implicitly motivated by the same objectives [24]. In both cases, the organizations, and especially the ITC, make a significant contribution to the creation of knowledge.

European ceramic tile clusters (Italian and Spanish) could be characterized as social networks in transition between the old and new social network taxonomies [26], with technical knowledge of migration from tacit to explicit and from mature to new and systemic taxonomies. These clusters are oriented towards radically new products, with external sources of innovations# and a knowledge base centered on the search for relational and cognitive networks. In these clusters, high-tech and non-high-tech subsectors are highly symbiotic [27], and they show sustainable endurance. As Figure 1 shows, these clusters display an active profile, considering productivity and value-added ratios and the fact of their much higher salary costs.

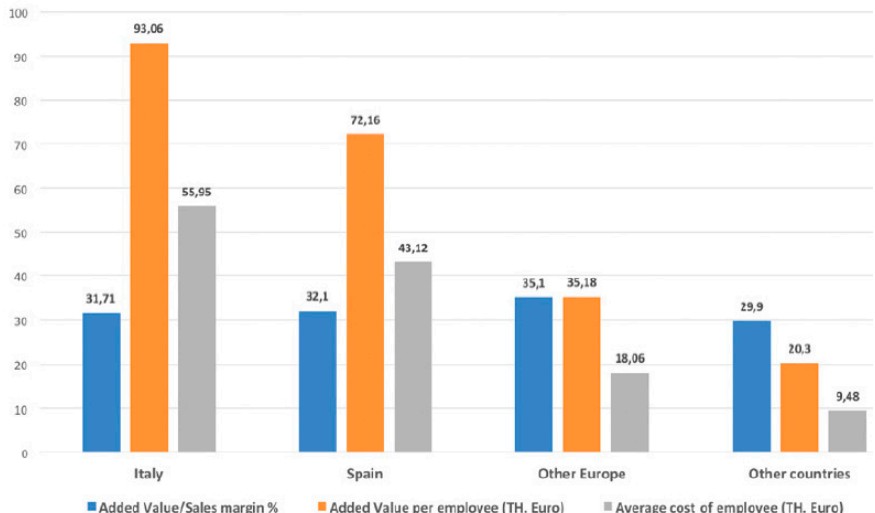

**Figure 1.** Productivity and added value of Italian and Spanish companies versus their worldwide competitors. Referenced from [28] with permission.

These clusters can be considered as being in transition from a typical industry-focused cluster, where specialization has been concentrated in specific subsectors that are capable of reinventing their value chain paradigms through innovation and internationalization strategies (i.e., equipment and plant engineering in Italy and glazing and pigments in Spain) [29]. This transition has been influenced by changes in technology and a globalization movement connecting the various ceramic tile clusters and configuring a global value chain.

There are examples of how clustering has enabled a deepening in the local knowledge base of traditional industries. Additionally, it has facilitated the establishment of a broader set of knowledge related to materials and machinery suppliers. This development is the ability of clusters firms to collectively transform low- and medium-tech industries into tacit knowledge-intensive industries, and to internalize this competitive advantage within the cluster. It implies a process of continuous learning with the support of intercluster linkages and tacit knowledge transfer supported by the financial support of regional and government agents [30].

It, thus, appears that networks (composed of formal and informal ties) become the locus of innovation in clusters, as the generation of knowledge is a crucial factor in improving the firms' competitive position [31]. Heterogeneity in the portfolio of collaborators allows firms to learn from a comprehensive stock of knowledge [32].

## 4. Radical Innovation and Disruptive Technologies

We will try to analyze and link the concepts of radical or disruptive innovation. The idea of radical innovation draws from the Schumpeterian concept of creative destruction, which was defined in the late 1970s. Abernathy and Utterback [33] introduced the concepts of incremental versus radical innovation. These authors [34] classified innovation according to market knowledge and technological capabilities. They distinguished between revolutionary innovation, which makes technological capabilities obsolete but preserves market knowledge; and architectural innovation, where both technological and market capabilities become obsolete. Radical innovations are fundamental changes that represent revolutionary changes in technology. These represent definite departures from existing practice [35] and entail substantial changes and discontinuities [36,37].

A theoretical model of innovation should consider three kinds of variables: (a) the distribution of knowledge, including the depth and diversity of knowledge and extent of exposure to information obtained from external sources; (b) attitudes of the organization's management and the value they place on change; and (c) organizational structure, including the effects of centralization upon adoption behavior [38]. However, "innovation includes both major and minor changes" [39]. An extremely

major change is called a radical innovation, although it is interpreted as radical in a technological sense. It is usually the case that in the early stages of a new industry, radical product innovation is the prevalent mode of innovation. However, it often has little if any economic impact, because product design is still in flux and the market is uncertain.

Radical innovation is a significant kind of change that represents a new technological paradigm. This implies that the codes developed to communicate changing technology will become inadequate [40]. Radical change creates a high degree of uncertainty in organizations and industry. It also sweeps away significant parts of previous investments in technical skills and knowledge, designs, production techniques, plants, and equipment. The supply side does not necessarily delimit the change. It comes from a change on the demand side and in organizational or institutional structures. The standard definition of radical innovation incorporates four primary dimensions: technology uncertainty, technology inexperience, business inexperience, and technology costs [41].

Radical innovation has also been associated with discontinuities. Technology evolves through periods of incremental change punctuated by breakthroughs that either destroy or enhance a firm's competences in an industry. In general, competence-destroying discontinuities are initiated by new firms, while those enhancing competence are initiated by existing firms [37]. Leading companies stay close to their customer demands. Thus, technology changes that damage established firms are not radically new or challenging from a technological view, but either: (a) present different performance attributes, previously not valued or known by existing customers; or (b) the value attributes may improve at such a rapid rate that the new technologies can invade those markets [42].

Other authors have analyzed how radical innovation is conducive to breakthroughs in their innovation context [43]. Recent theories postulate that market demands offer complementary explanations for technology disruptions [44]. Furthermore, it has been suggested that the ecosystem will influence technology substitution and disruption [45].

Disruptive technology was analyzed in two seminal articles [42,46]. These authors defined disruptive technologies as those which "bring to a market new value propositions". They distinguish disruptive technologies from sustaining technologies. While the latter improves the performance of established products along performance dimensions valued by mainstream customers, the former underperform in these segments but offer other features valued by fringe and new customers.

It has been argued that new products based on disruptive technologies have different attribute sets than existing products and only satisfy a niche market segment of performance dimensions, on which the disruptive technology excels [47]. Eventually, disruptive technology changes the basis of competition by changing the performance metrics with which firms compete. Moreover, it has been stressed that products based on radical, disruptive innovation have higher performance in dimensions valued by foreign or emerging markets and disturb prevailing consumer habits [48]. Thus, disruptive innovations are disruptive to both producers and consumers and usually are not demand-driven, but rather a result of supply push processes originated by entrepreneurs rather than incumbent firms [47].

An appealing approach to the development of disruptive technology was postulated [49], outlining what Christensen [46] had already analyzed, the role of the incumbents' lack of vision of their market and their fear of cannibalizing their existing assets in serving the market. This author challenges the theory of S curves as a base for technology prediction and firm strategy, and states that this must be applied to the analysis of the development of new disruptive technologies. He states that not only do small, new entrants introduce disruptive technologies, but also that large and incumbent firms can develop them. However, incumbents do not generally consider investing in disruptive technologies to be a rational financial decision, and here the lead users' roles in the disruptive innovation or technology adoption process are crucial [49,50]. In the same line, Chesbrough points out that the role of business leaders in the implementation of disruptive innovations is significant [51].

The factors that inhibit the development of disruptive innovation must also be considered. The organizational rigidities of many firms and the existence of dominant designs associated with successful concepts and risk-averse cultures are sometimes associated with the fear of cannibalizing

existing products or technologies, a stable status quo, the technological trajectory, the lack of infrastructure (knowledge, technologies, industries) required for disruptive innovation, and above all, the mindset barrier, mainly consisting of the inability to unlearn existing practices and paradigms [52].

The concept of *encroachment* must be considered here, describing the gradual advance of new disruptive technology beyond usual or acceptable limits, from the low end of their existing market towards the higher end, opening up new fringe markets that later become broader mass markets [52]. In this way, and as pointed out by Adner [44], the heterogeneity of customers' demands brings about a converging view of lead users [44,49].

We could, at this stage, conclude by proposing a new definition of disruptive innovation as being "a force that changes the paradigms or mindset of the industry related to the technology, its value propositions, and consequently, its application aspects".

The literature on the emergence of radical innovation in clusters, however, presents an unresolved controversy to which we attempt to contribute. The rapid and easy diffusion of tacit knowledge that occurs in agglomerations and clusters, on the one hand, may result in the generation of radical new ideas [11]. This perspective emphasizes the high interfirm labor mobility, the strong presence of lead users, as well as social issues that drive and reinforce formation of interfirm and interpersonal ties and facilitate sharing of conventional paradigms, among others. In short, this theory posits that clusters are innovation-enabling environments that promote the emergence of new radical technologies and innovations. Inertia, institutional uniformity, and lock-in, on the other hand, indicate that agglomerations and clusters are primarily aimed at generating incremental innovation, and that frequently lock-in and decline follow from this [53]. An excessively narrow search focusing mainly on local technologies and knowledge may turn clusters into spaces where creative destruction occurs with difficulty [54]. Furthermore, the excessive focus on local focal industry knowledge brings lock-in in agglomerations and prevents change from taking place [55,56]. Therefore, which perspective should we consider?

## 5. Case Study: The Kerajet Development

### 5.1. Technology Context

As has been mentioned, the development of mechanical equipment technology has been led by Italian firms in ceramic tile clusters in Europe and worldwide. During the second half of the 1980s, a technology paradigm change, namely the development of the single firing process [19,55], opened up opportunities for Spanish glazing, pigment, and enamel producers, which have led totechnology developments worldwide since that time. As a consequence, this cross-fertilization process has allowed Italian and Spanish ceramic tile clusters to lead ceramic tile technology over the past 30 years [20,25,57].

Until 1994, the decorating process in the ceramic tile sector was mainly based on screen printing technology utilizing flat or cylinder screens. Screen printing is a form of stenciling where the ink is forced through the clear elements of the stencil onto the substrate to be printed, in this case glazed slabs of crude tiles. Artwork has to be processed photographically in order to produce the screens, requiring manual manipulation. This process was burdensome, requiring the production of a set of screens (four or six) for each design and replacing them when the wear was excessive. Moreover, it required a large batch series, which produced large material stocks if tone changes had to be avoided. In 1994, the Italian company System released the Rotocolor machine, which replaced the screens with laser-engraved polyethene rollers, which transferred the design color patterns to the tiles. As in screen printing, four or six cylinders are needed for the primary design colors. Although this technique was a significant improvement, it did not solve all the design reproduction problems and implied the need for specialized technicians that could manage the production process. Furthermore, it still required electronic engraving of the rollers and large production batches. Furthermore, the design transfer process was arduous, lengthy, and costly. Additional problems existed relating to machine control, process instability, and the difficulties in reproducing a specific design (and a tile size limitation

of 1400 × 1300 mm.). Some competitors copied this design, causing several legal litigations [58], as the Rotocolor proof was becoming the dominant technology. Nevertheless, by the end of the 1990s. this technology had been adopted in 10%–15% of ceramic tile-producing plants.

### 5.2. The Development of a Breakthrough Innovation

In 1998, a local Spanish computer entrepreneur engineer with extensive experience in the ceramic tile industry, along with a chemist working in a leading glaze and pigments multinational, began to discuss new possibilities for decorating tile ceramics based on digital technologies. After several garage-type trials, in 1999, with public funding from CDTI (National Center for Technology Development), they developed a prototype based on inkjet printing (see Figure 2). The initial prototype proved its feasibility, and this led to the founding of an entrepreneurial firm, Kerajet. Using a design consisting of multiple inkjet head systems, control hardware, software design transmission, and inkjet handling subsystems, they presented their first industrial prototype to the CEVISAMA exhibition in March 2000, winning the First Alpha Gold Innovation Prize at the exhibition. Kerajet also presented two PCT (Patent Cooperation Treaty) patent applications.

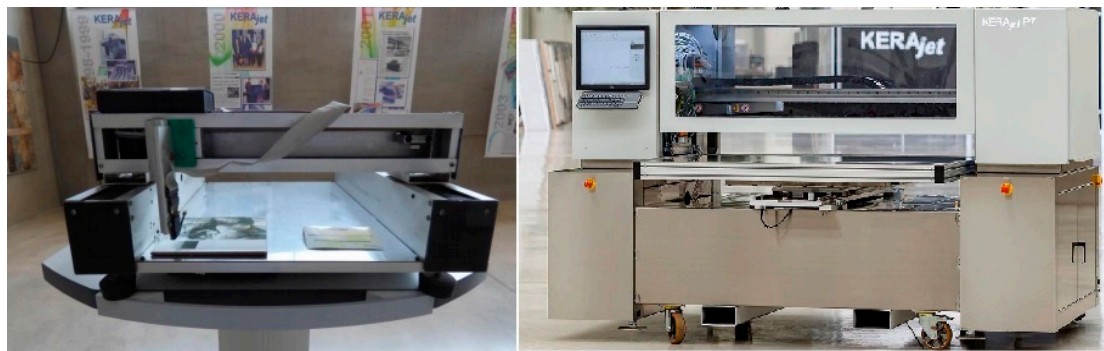

**Figure 2.** First Kerajet 1999 prototype and latest P7 model 2019 (source Kerajet).

At this early stage, financial support from the glazing and pigment leading producer, Ferro (a large MNE) was crucial. It was agreed that Kerajet would develop electronics, software applications, and the decorating machine, while the glazing MNE would focus on the ink's development.

Before proceeding to analyzing the case further, the systemic character of the new technology must be emphasized. In order to assemble an inkjet printing device, the developers had to consider four basic subsystems: (a) inkjet print heads, (b) inks or colors to decorate the tiles, (c) the mechanical hardware to move the tile and coordinate the print head function, and (d) the software to transfer the artwork to the printing system and control the process. All four subsystems needed to be coordinated in order to achieve the minimum performance of the new technology. In the subsequent developments, some of these subsystems (c,d) had a continuous evolution following a steady state of art progress, while others (a,b) followed a punctuated evolution. As discussed below, this was a critical question in the technology's evolution and subsequent adoption by customers.

Finally, the way in which inkjet technology changed the ceramic tile decoration process paradigm must be considered, not only considering the advantages of the new technology, but as a direct alternative to process artwork without physical intermediate manipulation (screens or rollers). This represented a complete breakthrough in the decoration process, which until then had been considered a cooking craft process [58], and which now became a digitized process.

### 5.3. A Long Road for Technology Adoption

Due to the newness of the technology and some of its technical drawbacks, such as the difficulties encountered in adapting existing ceramic colors (based on inorganic water-soluble pigments) to print heads developed for other applications, a major problem occurred with ceramic tile producers' adoption

of the new technology. This caused a considerable dilemma for the innovators [46]. The decision involved adopting, on one hand, a new form of technology that meant changing the existing craft production culture towards a digital computing environment, and on the other hand, a technology whose advantages were not apparently clear and which had been invented by a Spanish firm (meaning Italy, as the leader in mechanical ceramic tile equipment, was reluctant to adopt technology developed in other countries).

The role of the final user of tile ceramics was not revealed until later in the development process of the technology. Cutting edge designs and applications were shown to customers and the possibilities of the new technology to imitate and even improve patterns of natural materials, such as marble or wood, had an impact on their demand for tile ceramics decorated with inkjet technology. However, only three or four tile producers with capacities in advanced production technologies really understood the implications of the new technology and were committed to it.

During the dormant phase of inkjet technology development (2000–2004), few lead users (i.e., Zirconio, El Molino) adopted the technology enthusiastically, and their profiles were varied. One of these lead users was a medium-sized firm (300 employees), an exporter to Germany and England which specialized in cutting edge designs, who acquired four inkjet printers and contributed to a new generation of Kerajet printers with their comments and suggestions. This firm was the first in the Spanish cluster to incorporate data acquisition and control of the decoration process, and to be capable of envisioning the competences of the new technology. In a personal interview, their plant engineer showed us an excel table demonstrating the enormous savings of the inkjet alternative. The rest of the early inkjet adopters were amazingly small companies (60–70 employees). In an interview, one plant foreman explained to us how the new technology that they had struggled to learn involved having an ability to cope with short-batch deliveries and cut down inventories to satisfy their niche customers. However, the problem associated with this was that lead users believed they were developing differentiating competences and avoided disseminating their knowledge throughout the cluster, while lead producers tried the technology but rejected it, since it did not meet the needs of their mainstream customers, and their rejection was disseminated. One of the largest tile ceramic producers pointed out "when our Italian competitors buy it, we will do it as well".

These two occurrences, namely digital inkjet technology development and its slow adoption by ceramic tile producers, took place at the same time. The fact of the mutual influence and communication between the Italian and Spanish ceramic tile clusters was essential to the initial difficulties with the new technology, and its conclusion and eventual success.

*5.4. Technology Development and Success*

As has been mentioned, the Kerajet team had to cope with two technical problems that required knowledge from outside the cluster. Having selected the technology (piezo electric drop-on-demand), they had to select a print head producer who could adapt to ceramic tile decorations. It has to be emphasized that this effort was carried out entirely by Kerajet by travelling outside of Spain to the United Kingdom and Japan and meeting leading printhead producers, since Asian digital printing producers had only marketing infrastructure in Spain. After various trials and mishaps with XAAR (this firm was reluctant to invest in tile ceramics since they thought in 2005 that the foreseeable market did not justify the needed R&D investment and waited until three years later to advance technologically in this sector), the technology leader in print heads, they reached an agreement with Seiko to develop print heads specifically designed for ceramic tile applications (Seiko took care of the piezoelectric system and Kerajet handled the microelectronics). This cooperation lasted from 2002 to 2009. Additionally, Kerajet had to equip a laboratory to pilot test print heads for cleaning, drop analysis, and other tasks, and reached agreements to develop software with external research labs.

In relation to the inks required for the application, it soon became clear for the entrepreneurial team that the state-of-the-art pigment technology (based on inorganic soluble salts) was not compatible with the print heads required for inkjet technology. In this area, there were two apparent problems.

Kerajet's MNE partner was reluctant to invest heavily in the new technology and there were technology barriers for producing the new organic pigments due to the fact that the pigment size, which required avoiding obstructing the print heads, could not be successfully handled with ceramic tile milling technology. Then, Kerajet built a small lab in their premises to pilot the new ink's development, utilizing nanotechnology micromills. They also tested new organic solvents to produce the required inks. Thus, by 2004 they had already developed relevant advances in the technology, using print heads and inks that solved the most acute problems associated with ceramic tile inkjet decoration.

It must be emphasized that incumbent firms in the Italian mechanical equipment sector were reluctant to follow the new developments, in spite of them featuring in all relevant European ceramic tile fairs and exhibitions. These firms were slow to react to the new developments. It was not until 2007 that the leading firm System signed an agreement with Kerajet, with the intention of cannibalizing its own Rotocolor technology (Kerajet had a temporary agreement with System, inventor of Rotocolor, to integrate its technology into their system), and Sacmi registered its own patent with powder injection in 2008.

It can be concluded that by 2005, Kerajet was the leader and the pioneer in inkjet technology. Its printers had been recognized internationally at the Tecnargilla Fair (International Exhibition for Tile Ceramics Equipment) in September 2004 in Rimini, Italy. The firm developed and commercialized not only conveyor inkjet printers, but also a large flat-bed printer with moving print heads.

### 5.5. The New Technology Becomes a Dominant Design

The mid 2000s marked the development of inkjet technology as a dominant design. The glaze and pigment leaders followed the path of Kerajet and started to develop and market new inks for inkjet technology after realizing that it had a much higher added value. Kerajet was challenged by new entrants, mostly from the pigment and glaze sector, which was searching for outside partners in the inkjet technology sector and had to face the first litigations for patent infringements. The first follower was a pigment producer (Torrecid), who partnered with Durst in 2005 to present the second inkjet printer on the market with organic pigments. This was later followed by Cretaprint, a small Rotocolor manufacturer in Spain, later acquired by EFI (Electronics For Imaging) in 2012.

Print head producers, led by XAAR, started to develop inkjet printheads adapted for tile decoration, and within five years, ceramic tile inkjet print heads had become a standardized product, with four international firms making up 99% of the market. Organic pigmented inks also became standard, and Spanish glaze and pigment producers incorporated them into their catalogues. The two initial inkjet printer manufacturers (Kerajet and EFI-Cretaprint, based in the Spanish cluster) dominated the market with a 70% share until 2015 [59]. The remnants belonged to three or four manufacturers, including two Italian equipment producers. Durst has initiated their operations in Spain and later moved to Italy [60]. Therefore, the Spanish cluster dominated the technology in the initial years.

It must be pointed out that the initial lack of infrastructure inhibiting the development and dissemination of inkjet innovation, such as the lack of software competency, microelectronic suppliers in the cluster, print head technology suitable for the ceramic tile applications, and the lack of computer-trained operators, were surmounted by the visionary effort of the entrepreneurs who initiated change.

These facts confirm the effect of dominant design on encouraging standardization, so that production or other complementary economies could be sought [61]. The new technology offered incredible image resolution, line speeds, and productivity, as well as the potential for cutting edge designs that were not thought of previously. This has now been recognized as a leading competitive technology and the leading inkjet equipment manufacturers are now overflowing with numerous orders. New printer models were developed with increased numbers of attributes and improved specifications. At the end of 2008, there were more than 500 ceramic tile process lines equipped with inkjet machines. This underlines the role of Kerajet as a technological gatekeeper in the cluster [55].

Figure 3 shows the worldwide evolution of inkjet printer sales, comparing the leading firm's sales (Kerajet) versus total printer sales during the period 2000–2016. The graph shows how during the initial years (2000–06), the pioneer firm (Kerajet) completely dominated the market until 2012, with an estimated 50%–60% market share. Later, it lost its leadership. Digital tile printing is nowadays a mature and dominant technology. There are twenty main producers of inkjet printers. The clear leaders are in Spain, namely Kerajet and EFI-Cretaprint. Five producers are located in Italy, namely Durst, System-Coesia, Tecnoferrari, Projecta, and Sacmi. Finally, there are five main Chinese manufacturers. On the printhead side, there are two main competitors, Fuji and Xaar, followed by Toshiba, Seiko, and Konica. The technology is open in this regard and most printers can work with all printheads.

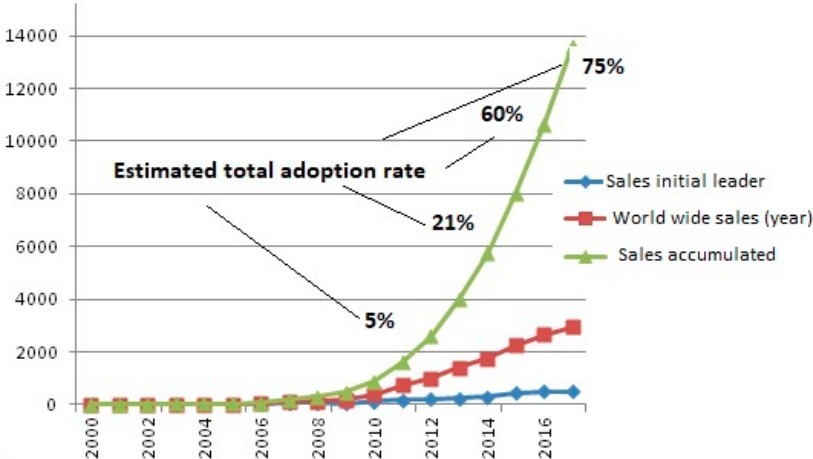

**Figure 3.** Sales curves of Inkjet technology. Estimated number of sold printers based on secondary information [62,63].

The historical sales evolution follows an exponential curve, and as the figure shows, the technology has reached a maturation phase. In 2014, the leading countries in inkjet adoption were Spain at 80% and Italy at 70%. As Italy is the industry leader in design, its lagging adoption figure may be interpreted as a consequence of the country's unwillingness to adopt foreign technology. This is the reason why Kerajet opened an office in Sassuolo in the center of the Italian cluster in 2007.

Thus, this case shows how disruptive innovation has been able to change a traditional industry based in an initial craft paradigm, relying on physical interfaces to decorate tile ceramics.

*5.6. Facilitating Elements in the Development of the Technology*

According to various authors, the following elements have been important in the final success of the Kerajet expansion. Some of these authors are classical writers of entrepreneurship literature, but we will concentrate on those who are essential to the elements involved, namely the cluster structure and the disruptive theory model. Additionally, these constitute elements were crucial in order to neutralize the inhibitors of disruptive innovation capabilities in the cluster, as already pointed out [64].

The first actors responsible for the project's success were the initial entrepreneurs. Their knowledge of various specialty areas in the Spanish and Italian clusters (equipment suppliers, tile producers, customers, pigment and glaze producers) combined with their skills (information and communication technologies, mechanical engineering, electronics, and chemistry) and their vision of the industry were the main drivers of the new project. Their vision was a necessary requirement to overcome the conservatism in innovating and applying new technologies to tile decoration processes, an area where craft was the dominant paradigm (for a view on the production technology approach of tile ceramic producers, see [57]). Kerajet acted as a focal firm and gatekeeper [65], as well as the technology

leader [47,51], by overcoming the district's lack of critical competences and bridging external knowledge when required, as was mentioned in the case of hardware and software for printheads.

The fact that Spanish incumbent actors from the equipment sector did not play a leading role in the development made it somehow feasible to innovate in that field [51]. However, some of them have played a follower role afterwards (Cretaprint). The ITC research center did play a relevant role by paying attention to the technology, disseminating it, and contributing by training technicians in seminars and workshops.

Next, we must consider the fact that the entrepreneurs established, through their own initiative, numerous links with various firms inside and outside the cluster, which opened the cluster to knowledge from other industries in an open innovation effect, thus developing the gatekeeping role. Additionally, research cooperation was carried out with two inkjet print-head manufacturers from Europe (XAAR) and Japan (Seiko). This led to the development of customized print-heads for usage in the ceramic tile field, and eventually the standardization of the application. The development of electronics and software for the control and management of the equipment was carried out in cooperation with various external university research centers and firms (i.e., the Polytechnic University of Madrid). Artwork software selection and training was essential for the transfer of designs to the operating line.

Though there was initial reluctance from pigment and glazing producers towards adopting a new technology that challenged the pigment status quo, a multinational firm, (FERRO), contributed equity and capital to the enormous investment initially required by the project. Later, cooperation with equipment suppliers in the pigment industry was fundamental in the development of process innovation for new pigment production. External knowledge in the micromilling processes provided by Kerajet was also essential during these phases. However, the leading position of Spanish enamel and pigment producers played a significant role here. Though Italian equipment manufacturers viewed the new technology as a threat to their main business areas, an effect of the innovators dilemma, the Italian inventor of Rotocolor, System, was a temporary partner in the project and contributed indirectly to the dissemination of the technology.

A fundamental role was played by lead users in the tile producer sector [50]. Three or four producers in Spain possessing outstanding production competencies made a commitment to the new technology, not only by the early acquisition of machines but by cooperating with numerous suggestions during the subsequent development of new models. At times, 80% of the changes in a new model were based on these lead users' comments [50]. Most of them built basic competences centered around their dominance of the new technology. Early in the development of inkjet technology they were aware of its cost advantages, and profited from its design and logistics improvements. When inkjet technology started to be popular, some lead users substituted almost all of their screen printing lines for digitized equipment. Kerajet made a marketing mistake by undertaking a global marketing initiative instead of concentrating on the innovating lead user producers. The standard tile ceramic producer at the time required a standard technology suited for their mainstream customer market and was not prepared to endure the learning curve that new technology requires. Word-of-mouth worked against the adoption of the new technology, since for lead users it meant a technical advantage, and they were not willing to disclose it. Thus, it meant a classical role of lead users as visionaries of need demands.

Regional and national innovation supporting agencies met and welcomed the project proposals with a positive attitude, and facilitated its development with generous subsidies and soft financing from the early start-up to the final refining stages. Their support was fundamental for the sustainability of the research project. Furthermore, their participation in a European research program supported the dissemination effort.

As was mentioned, the Spanish and Italian ceramic tile clusters were interconnected. It was assumed that all innovative efforts in the decoration process were led by Italian mechanical equipment manufacturers, while chemical innovation was led by Spanish glaze and pigment companies. On the other hand, the culture of the industry was conservative, and while the adoption of information technologies was acceptable in the marketing and distribution areas, it was still low on the production

line [4,58]. Therefore, the industry was passive and demanding towards the new digital decoration technology, although it observed its developments with interest. Worldwide industry exhibition fairs, such as CERSAI and TECNARGILLA in Italy, COVERINGS (Ceramic Tile and Natural Stone Industry North American Exhibition) in the United States, and CEVISAMA in Spain, demonstrated the progression of the technology during the 2000s. New equipment was exhibited and ceramic tile producers showed cutting edge designs imitating marble, along with natural stones and classical vintage decorations based on the new technology. Nevertheless, the transition of this disruptive technology from low to high market application was slow and took almost six years. Figure 4 illustrates and resumes the critical internal and external networking and partnering connections in the innovation process that led to the development and dissemination of the new inkjet technology.

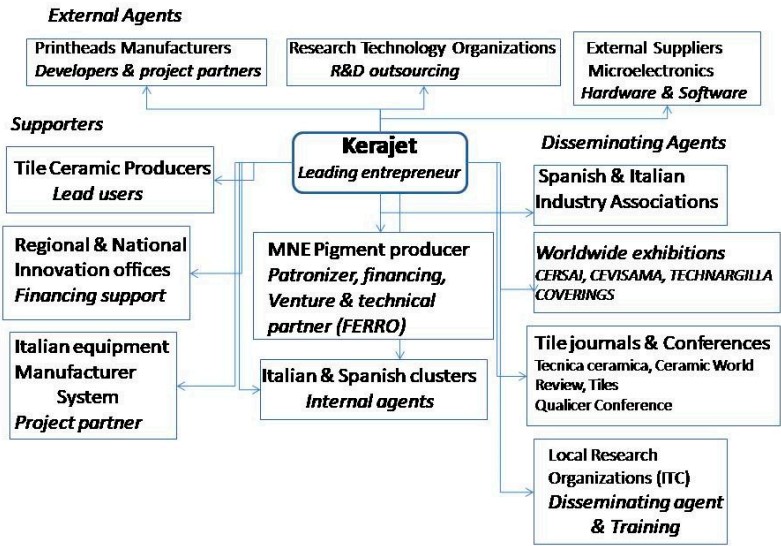

**Figure 4.** Facilitating elements in the technology development and diffusion process (authors). Abbreviations: ITC = Technological Ceramic Institute in Castellón.

Within the Italian and Spanish ceramic tile clusters, Kerajet was the agent of change as an expert of the trade. Its partnership and finance support from the MNE glaze and pigment producer FERRO was fundamental for funding the initial research pilots, which were estimated to cost more than three million Euros. This partner also provided the chemical knowledge required to develop the new ink paradigm throughout the project, although due to the companies risk aversion, it delayed certain phases of the project. Secondly, from the institutional side, national and local innovation agencies financially supported the successive phases of the development project. Thirdly, industry exhibition fairs (CERSAI, TECHNARGILLA, and CEVISAMA) favored a scenario for worldwide discussion and dissemination among competitors and customers. The comments published after each exhibition offer a track record of the evolution of inkjet from a disruptive technology to an accepted standard. The rest of the agents from both clusters also played a basic role. Lead users were critical as the early adopters and reviewers of the successive developments, while ITC contributed to disseminating the technology and training operators and technicians involved in with the product. Industry associations: ASEBEC (Spanish Association of Tile Ceramic Equipment Manufacturers), ACIMAC (Italian Association of Tile Ceramic Equipment Manufacturers), ASSOPIASTRELLE (Italian Tile Ceramic Producers Association), ASCER (Spanish Tile Ceramic Producers Association), ANFFEC (Spanish Fritt, Pigment and glazes Producers Association), and technical journals (e.g., Técnica Cerámica, Tile and Stone Journal, World Ceramic Review) sponsored workshops in Italy and Spain, where the applications were discussed and the technology information was disseminated worldwide. Incumbent firms in the equipment sector played active roles (i.e., System as a partner, Cretaprint as a follower) or passive ones as observers. Pigment and glaze producers facilitated the technology, either by being early followers

and competitors (Torrecid), or simply by being late adopters and facilitating the standardization of pigments for the new application.

Kerajet performed a fundamental role as a leader, gatekeeper, and acquirer of external knowledge. The formal and informal partnership with external print head manufacturers, Seiko (Japan) and XAAR (United Kingdom), was fundamental in developing print head technology totally suited for the ceramic tile applications. Hardware and software had to be selected, outsourced, or developed with the support of foreign agents or research laboratories in order for the state-of-the-art to progress. Pigment micromilling applications (i.e., Netzsch) solved the initial phases of creating organic pigments with the help of other external industries, such as pharmaceuticals or electronics.

This entrepreneurial firm demonstrated "the internal driving energy to generate and explore radical new ideas and concepts, to experiment with solutions for potential opportunity patterns... and to develop them into marketable and effective innovations, leveraging internal and external resources and competencies", following a definition of disruptive innovation capabilities [62].

## 6. Main Implications of the Results and Discussion

The case outlines the relevance of social capital in innovation dynamics in clusters [66]. On the other hand, the effects of partnership are often evident [32,67]. In this case, the cluster is a place in which social interaction, context, and shared cognitive schemes build the proper framework for learning and knowledge creation. It confirms the propositions of Belussi and Sedita [6] on internal and external knowledge acquisition. The case points out the importance of network diversity as a facilitator of radical innovation, as well as for the number of interacting contacts [68]. From this point of view, radical innovation in a cluster requires high levels of social innovation, bringing together capital, trust, mutual understanding, shared values, and at a very early stage, all the stakeholders who are critical in the innovation process [69].

The most remarkable suggestion that this case poses is an innovative pattern change, where the Italian mechanical firms used to lead all innovations in the mechanical engineering of the ceramic tile process lines, including limestone atomizing, slab presses, flat screen or rotogravure printing, glaze dosing, kiln firing, and packaging; while the Spanish glaze and pigment firms were the innovation leaders on the chemical side. These companies had roles as gate keepers in the cluster's social structure. Therefore, an outsider firm adopted the role of challenging the technology status quo. However, emerging novel technologies cannot achieve their potential under the systemic limitations imposed by existing structures, practices, and ways of thinking if the cluster or network is unable to cope with them. From another perspective, they must overcome the inhibitors of disruptive innovation capabilities [70].

However, as it has been mentioned, clusters, and especially those based in traditional industries, have usually been associated with incremental changes. A new paradigm must be contemplated then, regarding the function of cluster analysis in less stable and more complex environments—clusters as complex adaptive systems. These systems are composed of multiagent systems constituted by adaptive agents [71]. They (Kerajet in our case) are capable of coping with emergent properties, generating radical innovations, adapting to environment changes, restoring internal stability (coping with a radical new dominant design), and adapting to complex dynamics. This new cluster paradigm will allow us to understand how dynamic clusters can cope with uncertainty and change.

From a disruptive innovation point of view, it is necessary to ask which factors facilitated the development of technological changes that had such a worldwide impact. The prevailing mechanical engineering culture in the ceramic tile firms hindered the Italian mechanical firms from detecting and exploiting opportunities for information technologies in the decoration process, although oddly they had been incorporated in other process areas. The theory of disruptive innovation can contribute to explaining the change.

In principle, the first question posed by a visitor to a modern ceramic tile plant would be to ask why the decoration process area was so dirty and messy. These areas give an appalling impression,

given that the modern plants that were incorporated at the start of 2000s have automation and modern controls. Additionally, artificial vision forquality control exists in the rest of the plants (the development of artificial vision substituting for human control in tile ceramics is also worth analyzing). Also, the craft decoration processes thatwere followed impeded easy reproduction of certain color tones and required large batches and inventories (Italian equipment manufacturers had incorporated electronic controls in raw material preparation, press areas, packaging, and logistics. However, in workshops held in Italy and Spain during 2005, attendants sustained the position that "decoration will be a craft area for years"). Decoration operators needed to be specialists, and the production changes required, even with Rotocolor technology, involved long and nonproductive time lags. Considering this, it is difficult to understand the opposition that inkjet technology was met with during its early years. This status quo had become part of a strong mindset in a traditional industry such as tile ceramics [49,64]. Incumbents were content with the situation, since final customers were unaware of the situation, and large inventories, as visitors to Italian or Spanish clusters could observe, were part of the landscape [46]. It is, thus, understandable that only visionary agents outside the cluster, or within its boundaries but with a deep knowledge of the industry, could propose a disruptive change of the status quo. The cluster may drive the emergence of radical innovation, though mostly from the periphery and less from central actors [11]. Furthermore, these entrepreneurial agents brought external knowledge to the cluster (information technology, microelectronics, software, etc.), acting as gatekeepers of the new paradigm [65].

It is, thus, necessary to ask whether it would it have been possible for this development to have taken place outside the cluster. The answer is certainly negative. As has been discussed, the knowledge of the heterogeneous needs of the customers [44], the capability of identifying lead users who played a critical role by testing the initial pilots of the technology [50], and the vision of the future performance capabilities of the new technology [47] were fundamental elements for the success of the development. However, this bridging process had to be carried out within both the Italian and Spanish clusters in order to benefit from the interactions in both. The encroachment effect [52] or development from the low end of the existing market towards the higher end would not have been possible without the support (or initial co-operation) from the glaze and pigment sector, in spite of the leading role of Kerajet during the pigment paradigm transition. Likewise, the previous knowledge of the ceramic tile market of the two main print head leaders (XAAR and Seiko) was a convincing argument for Kerajet to bring them into the project in order to secure one of the key systems in the technology, and thus to introduce new knowledge and overcome the infrastructure problems [64].

Lastly, as shown in this case, this study does respond to the existing tension about the development, or lack of development, of radical innovation in clusters: as results point out, the cluster constitutes an innovation-enabling environment that triggers the formation of radical innovations. Thus, our findings align with those studies that state that the rapid circulation of tacit knowledge, the strength and specialization in some technologies, along with other cluster features (interfirm mobility, sharing knowledge, social issues, etc.) make the occurrence of radical innovation possible [11].

## 7. Conclusions

This paper illustrates how clusters provide an appropriate atmosphere to host breakthrough innovation, due to the presence of lead users, the sharing of common paradigms and knowledge, and the social bonds which hold together the different components (such as technology centers, associations, educational centers, related industries). In fact, medium-sized high-tech companies, such as in machinery in Italy and glazing in Spain, were the firms that fed the bulk commodity producers by following the innovation systems model [21]. In addition, and following the seminal contribution of Tallman et al. [72], the diffusion of new knowledge occurs within clusters, due to the fact that the architectural knowledge (i.e., the clusters' resources and capabilities) can easily reframe its structure and assimilate and absorb the new knowledge generated in the local community. The spread of this new knowledge to other territories will take more time, especially for the newcomers, due to

the lack of proper architectural knowledge or absorptive capacity [68,73] to accommodate the news paradigms generated in the developed world. The advantage of the specialized industrial approach of clusters has been useful in this case. The case also showed how incumbent firms in clusters exploit their specific local resources as well as their external knowledge in order to develop market opportunities, along with the role that suppliers' specializations plays in innovative developments.

A new consideration of clusters in a globalized economy and in more complex and competitive environments is necessary for the understanding of cluster dynamics. The way clusters interconnect thorough the value chain requires further analysis in order to understand how knowledge, as well as products, can flow through cross-clusters. This may help to redesign the GVC. The case study also points out that radical innovation is also possible in low- or medium-technology clusters, or in traditional clusters. However, the effect of the innovation dilemma may be stronger than in high-tech clusters, where the adoption of innovations is faster and follows a clearer path. Our results also confirm the findings of Robertson and Patel [27], showing that high-tech industries feed low-tech ones with knowledge. However, the main contributions of this paper include explaining how radical changes can be triggered in traditional clusters by disruptive innovations; how these can take place by overcoming inhibiting factors; and how disruptive innovation encroachment processes can take place because of the closeness within the clusters and the role of various actors in the technology diffusion, such as customers, incumbent competitors, and other agents in the value chain.

Regarding the lead users framework, this contributed strongly to the development and adoption phases of the technology in the early part of the project. Moreover, academic literature had not sufficiently examined the lead role of users in clusters. Their proximity and embeddedness facilitates their identification and contributions. In this case, they had a special profile, since they were not all lead ceramic tile producers, but often had a more technical leadership in the production side.

From a policymaking perspective, from this case we have learned that clusters create a good environment and framework for innovation, creation, and diffusion. In addition, it has been shown that the presence of medium-high or high-tech sectors in traditional low-tech clusters is vital. Thus, policymakers should encourage the attraction of national and foreign capital to value-adding activities, which can complement and reinforce local traditional industries to secure the connection of local clusters within the GVC.

**Author Contributions:** J.A.-G. the disruptive theoretical framework while J.L.H.-O. contributed the cluster approach. Both authors carried out the interviews while J.A.-G. took care of the case study. J.A.-G. wrote the paper and J.L.H.-O. revised it.

**Funding:** This research received no external funding.

**Conflicts of Interest:** The authors declare no conflict of interest.

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
