# Peer review of "Disruptive Innovation in Traditional Clusters: The Case of the Kerajet Ceramic Tile Cluster in Spain"

_applsci, doi:10.3390/app9245513_

Round 1

Reviewer 1 Report

An interesting question - are clusters 'better' at incremental than radical innovation - and an interesting case study of a cluster that has featured in the classic literature on clusters - ceramic tiles in in Italy and Spain.

What should be sharpened, is the clarity of the research approach. What exactly is the question or hypothesis to be tested here, in what way can a case study shed light on it, and what are the ley data points/observations that lead the authors to draw conclusions in one way or the other. The article so far provides an interesting narrative and shows clear in-detail knowledge about what happened in the cluster. But it is relatively weak on the scientific approach and lessons to be drawn.

One question that could be addressed more is whether relationships with related clusters (or previously unrelated clusters in the same location) have contributed to the disruptive innovation (while they play no role in incremental innovation). Did, for example, the presence of Asian consumer electronics firms in Spain, including in printing, play a role? Seems in any case that here it is more on the supply side where the asnwer lies, given that there is no indication that differences in the markets served explain why Spain was faster in this new way of doing things.

Overall an article that will be a useful contribution to our knowledge of clusters and innovation.

Author Response

An interesting question - are clusters 'better' at incremental than radical innovation - and an interesting case study of a cluster that has featured in the classic literature on clusters - ceramic tiles in in Italy and Spain.

We have expanded this question by citing references dealing with the subject and discussing it in the conclusions

What should be sharpened, is the clarity of the research approach. What exactly is the question or hypothesis to be tested here, in what way can a case study shed light on it, and what are the ley data points/observations that lead the authors to draw conclusions in one way or the other. The article so far provides an interesting narrative and shows clear in-detail knowledge about what happened in the cluster. But it is relatively weak on the scientific approach and lessons to be drawn.

We have expanded the introduction section and expanding the research question paragraph 

One question that could be addressed more is whether relationships with related clusters (or previously unrelated clusters in the same location) have contributed to the disruptive innovation (while they play no role in incremental innovation). Did, for example, the presence of Asian consumer electronics firms in Spain, including in printing, play a role? Seems in any case that here it is more on the supply side where the asnwer lies, given that there is no indication that differences in the markets served explain why Spain was faster in this new way of doing things.

In the case description and discussion, we have detailed our recollection of this subject and clarify the approach of the entrepreneurs in this question.

Overall an article that will be a useful contribution to our knowledge of clusters and innovation.

Reviewer 2 Report

The paper is about disruptive innovation in clusters. It provides a theoretical discussion over the topic, plus an empirical analysis through a case study: the ceramic tile cluster of Kerajet, in Spain.

Overall I enjoyed reading the paper, which contributes to the literature on modalities of innovation in clusters. Nevertheless, some issues must be resolved before publication. Please consider the following comments during the review process.

I suggest changing the title, at the moment is very long and not really to the point. I'd keep something like "Disruptive innovation in traditional clusters: the case of the Kerajet ceramic tile cluster in Spain." The cluster life cycle does not seem to be a relevant framework for the interpretation of the results, therefore there is no need to claim in the title that clusters are mature, but still I believe that is important to specify that the clusters under scrutiny operate in traditional industries. In the introduction the authors present an overview of contributions that focused on knowledge dynamics and communities of practice in clusters. I reccomend reading Belussi and Sedita (2012), who deepened the theoretical understanding of learning processes in industrial districts by analysing the emergent and deliberate structures that favour knowledge transfer at the local and distance level. In their framework, communities of practice and business networks are two relevant mechanisms for knowledge transfer. When considering the importance of division of labor between clusters, again, I feel some references are missing, Belussi ans Sedita (2008), among the others. The paper is really about clusters, therefore I do not understand why section 3 refers to ceramic tile industry. I'd better describe the ceramic tile production in Europe at first, and then I'd move to ceramic tile clusters, where Italian and Spanish firms are main characters in the story. Consequently, I suggest dividing the sections differently, for instance: 3. The ceramic tile production in Europe; 3.1. An overview - where the authors may introduce the specificities of the ceramic tiles production and the topic of the international division of labor between different clusters; 3.2 The role of clusters - where the authors might focus on the interplay between Spanish and Italian clusters. 4. Radical innovation and disruptive technologies.

References

Fiorenza Belussi & Silvia R. Sedita (2012) Industrial Districts as Open Learning Systems: Combining Emergent and Deliberate Knowledge Structures, Regional Studies, 46:2, 165-184, DOI: 10.1080/00343404.2010.497133   Belussi, F., & Rita Sedita, S. (2008). The Symbiotic Division of Labour between Heterogeneous Districts in the Dutch and Italian Horticultural Industry. Urban Studies, 45(13), 2715–2734. https://doi.org/10.1177/0042098008098202

Author Response

I suggest changing the title, at the moment is very long and not really to the point. I'd keep something like "Disruptive innovation in traditional clusters: the case of the Kerajet ceramic tile cluster in Spain." The cluster life cycle does not seem to be a relevant framework for the interpretation of the results, therefore there is no need to claim in the title that clusters are mature, but still I believe that it is important to specify that the clusters under scrutiny operate in traditional industries.

Thank you for your suggestion, we followed your advice and changed the manuscript title accordingly.

In the introduction the authors present an overview of contributions that focused on knowledge dynamics and communities of practice in clusters. I reccomend reading Belussi and Sedita (2012), who deepened the theoretical understanding of learning processes in industrial districts by analyzing the emergent and deliberate structures that favor knowledge transfer at the local and distance level. In their framework, communities of practice and business networks are two relevant mechanisms for knowledge transfer. When considering the importance of the division of labor between clusters, again, I feel some references are missing, Belussi and Sedita (2008), among the others.

We have expanded the references including both suggested articles in our analysis. We have also included other references relevant to cluster learning and paths to acquiring knowledge. Articles discussing whether clusters promote radical innovation have also been included.

The paper is really about clusters, therefore I do not understand why section 3 refers to ceramic tile industry. I'd better describe the ceramic tile production in Europe at first, and then I'd move to ceramic tile clusters, where Italian and Spanish firms are main characters in the story. Consequently, I suggest dividing the sections differently, for instance: 3. The ceramic tile production in Europe; 3.1. An overview - where the authors may introduce the specificities of the ceramic tiles production and the topic of the international division of labor between different clusters; 3.2 The role of clusters - where the authors might focus on the interplay between Spanish and Italian clusters. 4. Radical innovation and disruptive technologies.

Thank you for your suggestion which we found quite adequate, We have followed your advice and reorganize the article accordingly.